# Carbon Nanotube-Based Electrochemical Biosensor for Label-Free Protein Detection

**DOI:** 10.3390/bios9040144

**Published:** 2019-12-17

**Authors:** Jesslyn Janssen, Mike Lambeta, Paul White, Ahmad Byagowi

**Affiliations:** 1Department of Biological Sciences, University of Manitoba, Winnipeg, MB R3T 2N2, Canada; 2Department of Physics and Astronomy, University of Manitoba, Winnipeg, MB R3T 2N2, Canada; mike@lambeta.net; 3Department of Electrical and Computer Engineering, University of Manitoba, Winnipeg, MB R3T 2N2, Canada; pauljwhite@gmail.com (P.W.); ahmadexp@gmail.com (A.B.)

**Keywords:** electrochemical biosensors, SWCNT, point-of-care diagnostics, label-free biosensors, ELISA, carbon nanotubes, bovine serum albumin

## Abstract

There is a growing need for biosensors that are capable of efficiently and rapidly quantifying protein biomarkers, both in the biological research and clinical setting. While accurate methods for protein quantification exist, the current assays involve sophisticated techniques, take long to administer and often require highly trained personnel for execution and analysis. Herein, we explore the development of a label-free biosensor for the detection and quantification of a standard protein. The developed biosensors comprise carbon nanotubes (CNTs), a specific antibody and cellulose filtration paper. The change in electrical resistance of the CNT-based biosensor system was used to sense a standard protein, bovine serum albumin (BSA) as a proof-of-concept. The developed biosensors were found to have a limit of detection of 2.89 ng/mL, which is comparable to the performance of the typical ELISA method for BSA quantification. Additionally, the newly developed method takes no longer than 10 min to perform, greatly reducing the time of analysis compared to the traditional ELISA technique. Overall, we present a versatile, affordable, simplified and rapid biosensor device capable of providing great benefit to both biological research and clinical diagnostics.

## 1. Introduction

Recently, the need for affordable, rapid and simplified diagnostic devices has been growing, especially in third world countries. Many diseases such as cancer and infectious disease require an early-stage diagnosis, when treatment options can be most effective. It is also suspected that drug discovery may be accelerated for incurable diseases such as Alzheimer’s disease where the inability to diagnose the disease at its earliest stages may be leading to ineffective drug trials which are usually administered to patients that possess irreversible, late-stage pathology. In the case of breast cancer, 5-year survival rates are much lower for developing countries such as Gambia (12.0%) and Algeria (38.8%) in comparison to the United States of America (83.9%) and Sweden (82.0%) [1]. It is believed that lower survival rates in developing countries are due to diagnosis in the advanced stages of disease and access barriers to medical care [2,3,4]. In the case of infectious disease, just under 1 million people die from malaria, 4.3 million people from acute respiratory infections and 2.9 million from enteric infections every year [5]. More than 90% of the deaths due to infectious disease occur in developing countries [5]. Affordable, rapid and simplified diagnostics are critical for combatting these diseases, yet most diagnostic methods are inaccessible to those who need them most.

Currently, immunological assays such as enzyme-linked immunosorbent assay (ELISA) for the detection of biomarkers in bodily fluids are sensitive and provide accurate results. However, the current assays involve sophisticated techniques and require highly trained personnel for execution and analysis. It may take days to weeks for patients to obtain results from the current immunoassay technologies. ELISA also requires large amounts of samples and due to its label-based approach, expensive and specialized reagents are needed, which prevents the use of these assays in resource-poor environments [6,7]. Therefore, the need for a rapid, inexpensive and simplified diagnostic device is still unmet.

Biosensing technology is a promising alternative, owing to its potential for rapid, simple, sensitive, low-cost and portable detection [8]. Biosensors are devices that comprise a biological component for recognition, and a physiochemical detector component for transduction [9]. The transduction component can be optical, electrochemical, piezoelectric, magnetic or calorimetric [9]. The biological recognition component can be developed using enzymes, antibodies, cells, tissues, peptides, nucleic acids and aptamers [10,11,12,13,14]. In regard to biosensing technology for protein detection, there is an interest in label-free biosensors as they require only a single recognition element, leading to a simplified design and a reduction in reagent costs and assay time [15]. Optical interferometry [16], fiber optic surface plasmon resonance [17,18,19,20], piezoelectric [21,22,23,24] and electrochemical [25,26,27,28] label-free methods have been developed to overcome the limitations of label-based biosensing technologies. In particular, electrochemical label-free methods are most promising with regard to high sensitivity, lower detection limit, lower response time, cost-effectiveness, miniaturization, simplification and portability [29,30]. Electrochemical measurement methods are most suitable for mass fabrication and have played a crucial role in the transition towards simplified point-of-care diagnostics [11]. This shift has been evident through the market domination of self-testing glucose strips, based on screen-printed enzyme electrodes, coupled to pocket-sized amperometric meters for diabetes over the past two decades [31]. Although various research efforts have been made, the development of a simplified, affordable, easy-to-use and highly performant protein detection system for point-of-care analysis remains a challenge.

Electrochemical biosensors that utilize nanomaterials such as carbon nanotubes (CNTs) for improved sensitivity and response time are potential candidates for point-of-care protein detection. CNTs are hollow, cylindrical molecules consisting of a hexagonal arrangement of hybridized carbon atoms, one or more walls and a nanometer scale diameter [32]. Their well-ordered arrangement of carbon atoms is linked via sp^2^ bonds, making them the stiffest and strongest fibers known [32]. Depending on their number of walls, CNTs can be divided into single-walled carbon nanotubes (SWCNTs) and multi-walled carbon nanotubes. In particular, SWCNTs offer great promise for biosensing applications due to a unique combination of electrical, magnetic, optical, mechanical and chemical properties [32]. SWCNTs exhibit the simplest morphology and may be formed by rolling up a single graphene sheet. The large surface area of SWCNTs enables interactions with a large number of biological agents at the carbon nanotube surface, allowing for sensitive detection of biomolecular interactions [33]. Additionally, the electrical conductance of SWCNTs are sensitive to the environment and vary significantly with surface adsorption of various chemicals and biomolecules, making SWCNTs promising candidates for label-free biosensing [34,35,36].

Due to their unique characteristics, SWCNTs have been used in several label-free detection modalities. One such label-free modality includes electrical percolation-based biosensors. In electrical percolation, a long-range electrical connectivity is formed in randomly oriented and distribute systems of conductive elements, such as SWCNTs [37]. In these systems, the passage of current through the conductive network depends on the continuity of the network [36]. Electrical percolation-based biosensors often comprise a 3-dimensional carbon nanotube and antibody network [37]. In such biosensors, molecular interactions, such as the binding of antigens to antibodies, disrupt the network continuity causing an increase in the network’s electrical resistance [37].

This paper describes the development and performance of a label-free disposable biosensor based on cellulose paper impregnated with SWCNTs and antibodies for protein detection. The biosensors take advantage of the electrical percolation principle in order to facilitate label-free and simplified detection of protein biomarkers. Biosensors were developed for the measurement of BSA and comprised an antibody specific to BSA. These biosensors were developed for the detection and quantification of BSA because it is often used as a protein concentration standard in lab experiments. Additionally, targeting such a protein could be useful for determining whether our developed biosensors were capable of measuring a standard protein in solution as a proof-of-concept and whether our method could compete with other standard techniques such as ELISA. The data obtained in this work indicate that such biosensors may be utilized as a foundation for rapid, inexpensive and simplified diagnostics.

## 2. Materials and Methods

### 2.1. Material and Reagents

The poly(sodium 4-styrene sulfonate) (PSS) and antibodies specific to bovine serum albumin were obtained from Sigma Aldrich (St. Louis, MO, USA). SWCNTs were purchased from Cheap Tubes Inc. (Brattleboro, VT, USA) and used without purification. Filter paper discs were cut from Whatman cellulose filtration paper obtained from Sigma Aldrich (St. Louis, MO, USA). A bovine serum albumin stock solution bought from Thermo Fisher Scientific (Rockford, IL, USA) was used for the preparation of various testing concentrations of BSA. Ovalbumin stock solution was purchased from Sigma Aldrich (St. Louis, MO, USA). The BSA ELISA kit was purchased from Cygnus Technologies (Southport, NC, USA).

### 2.2. Biosensor Fabrication

SWCNTs and PSS were dispersed in distilled water to achieve a final concentration of 50 mg/mL. The SWCNT and PSS solution underwent bath sonication for 30 min prior to centrifugation for 20 min for the removal of aggregates. PSS was used to facilitate electrostatic adsorption for protein immobilization. 8 μL of BSA antibody was added to 5 mL of the PSS and SWCNT solution. The SWCNT-antibody solution was vortexed for 1 min and then drop-casted onto 3 mm × 3 mm cellulose paper disks. The disks were then placed onto a printed circuit board (PCB) electrode and freeze-dried under vacuum to prevent antibody denaturation. Biosensors were also developed without the use of BSA antibody, in order to control for non-specific BSA binding.

### 2.3. Sample Preparation

Various concentrations of BSA were prepared from a 10% BSA stock solution. 1 ng/mL, 5 ng/mL, 10 ng/mL, 20 ng/mL and 40 ng/mL BSA solutions were prepared. Ovalbumin standard solutions used in our control experiment were prepared from a 10% stock solution in a similar manner.

### 2.4. Measurements

5 µL of each BSA standard solution were placed onto a biosensor disk and the biosensors were left to dry. To determine the optimal temperature at which the biosensors should dry, or the immunoreaction should occur, the drying and subsequent measurements were carried out under various temperatures (4 °C, 23 °C, 37 °C and 50 °C). All electrical measurements were performed using a programmed Arduino Uno. The Arduino measurement device was programmed to measure electrical resistance and record the value at the transient plateau of resistance readings, before subtracting the recorded value from the initial measurement that was taken prior to the reaction, resulting in a final value for the change in resistance (see Appendix A for further details). Changes in resistance were correlated with the concentration of BSA. To control for non-specific binding, various concentrations of BSA were applied to biosensors developed without the use of the BSA antibody. Additionally, concentrations of a non-target analyte solution were applied to the developed biosensors. The standard solutions of BSA were analyzed using traditional ELISA for verification and comparison. All experiments were carried out five times before the average values were calculated. The detection limit was calculated using LOD = 3.3 SD/S, where LOD is the limit of detection, SD is the standard deviation and S is the slope of the resulting standard curve of the trials with BSA standard solutions.

## 3. Results and Discussion

### 3.1. Biosensor Design

Accurate and rapid quantification of protein biomarkers plays an important role in the diagnosis of numerous diseases. With the potential that carbon nanotube biosensors offer for rapid, simple, sensitive and portable diagnostics, we developed SWCNT-coated paper-based biosensors for the detection and quantification of a standard protein, bovine serum albumin. Similar to a finger-prick blood test, the developed SWCNT biosensors were designed for the quantification of a specific protein in a single drop (5 µL) of sample solution.

Only SWCNTs were used for the fabrication of the biosensors, due to their larger surface area. Cellulose paper discs were drop-casted with the SWCNT/PSS/antibody solution, 6 times in a row to achieve an electrical resistance in a sufficient 200–250 Ω range. The discs were placed onto a printed circuit board (PCB) electrode which was subsequently freeze dried under vacuum. One of the fabricated biosensors is depicted in Figure 1. All biosensors used during experimentation had SWCNT-coated discs that were firmly attached to the PCB substrate. Attaching the SWCNT-coated discs to the PCB electrodes in this way allowed us to avoid clamping electrodes that bend the flexible paper-based biosensors, causing resistance changes due to strain that interfere with analyte measurement.

### 3.2. Biosensor Testing with BSA

Biosensors were tested with increasing concentrations of BSA standard solutions in addition to control solutions containing a non-target analyte. 5 µL of each solution was tested to generate a curve with increasing BSA concentration. After 5 µL of a particular solution was placed onto a biosensor disc, the reaction was allowed to proceed for 5 min before a measurement was recorded. This time-frame was chosen in order to allow for antibody-antigen complex formation in addition to the absorption of the sample on the disc.

With the goal of developing a biosensor based on the electrical percolation principle in order to facilitate label-free and simplified detection of protein biomarkers, we suspected that the magnitude of change in the electrical resistance of the developed biosensors following the addition of a sample solution would show a positive correlation with the concentration of BSA. A detailed schematic of the suspected sensing mechanism of our electrical percolation-based biosensor system is depicted in Figure 2. As expected, our resulting curve shown in Figure 3 displays a positive correlation between BSA concentration and the magnitude of increased resistance. Based on these results, it is clear that the presence of the analyte, BSA, reduces the current through the biosensor, and hence, increases the resistance of the SWCNT-paper disc. Therefore, these results are in concordance with the mechanism of electrical percolation-based biosensors. In such biosensors, the formation of antibody-antigen complexes disrupts the continuity of a conductive network, causing an increase in the network’s resistance, which our results exhibit [37].

### 3.3. Temperature Optimization

The performance of electrochemical biosensors is often strongly influenced by the environmental conditions, such as the temperature at which the reaction proceeds. Temperature can influence biosensor performance by affecting various parameters such as the antibody-antigen reaction and SWCNT electron transfer. Thus, temperature is a variable that required optimization prior to performance assessment and comparisons with standard techniques such as ELISA. Accordingly, biosensor performance was tested with standard sample solutions at 4 °C, 23 °C, 37 °C and 50 °C (Figure 4). All of these temperatures are regularly deployed for the analysis of immunoreactions except for 50 °C, which was arbitrarily chosen to display a broad coverage of the effects of temperature. Between the temperatures that were tested, biosensors showed the most optimal performance at 37 °C. Beginning at 4 °C, the efficiency of the biosensors increased with temperature until 37 °C which is typically the most efficient temperature for antibody-antigen reactions. At 50 °C, efficiency of the biosensors was minimal again due to the denaturation of the antibodies. As 37 °C was found to be the optimal temperature for biosensor performance, we show a standard curve of the testing results under such conditions in Figure 3.

### 3.4. Analysis of Biosensor Performance and Controlled Experiments

With the optimal temperature determined, we can evaluate overall biosensor performance and compare the developed biosensors to standard methods such as ELISA. At large, the performance of the developed biosensors greatly exceeded our expectations. With an LOD (limit of detection) of 2.89 ng/mL, the developed biosensors could sufficiently respond to even minor changes in BSA concentrations. The developed biosensors were also found to be highly specific as the addition of a non-target analyte solution, ovalbumin, and the addition of BSA to biosensors that were developed without BSA antibody resulted in only a slight change in resistance (Figure 5). Additionally, when increasing concentrations of ovalbumin were applied to the SWCNT-discs, or when increasing concentrations of BSA were applied to SWCNT-discs that were developed without the use of BSA antibody, there was no systematic correlation between the increasing concentrations and the magnitude of change in resistance. The slight changes in resistance were most likely a result of minor changes in the SWCNT network due to interference by the non-target or BSA analyte. Additionally, the weight of the solution may have caused strain in the SWCNT-coated disk or a modest change in morphology, leading to a slight increase in resistance. We suspect that with higher quality SWCNTs or further optimization of environmental conditions, the performance and LOD can be improved.

Our resulting standard curve for our developed biosensors displayed excellent linearity, with R^2^ of 0.9989 (Figure 3). The curve is in the range of 0–40 ng/mL and even at 40 ng/mL, there is no deviation from linearity. We suspect that the biosensor’s constituent antibodies are not fully saturated, and the biosensor may be capable of responding to more concentrated BSA solutions.

### 3.5. Analysis and Comparison with ELISA and Recently Developed Methods

We generated a standard curve with our prepared BSA concentrations using the traditional ELISA method in order to verify that our BSA test solutions were of the correct concentration (Figure 6). To generate this standard curve, we deployed a BSA ELISA kit that had an LOD of 6.26 ng/mL and a detection range of 6.25–400 ng/mL. As ELISA is a standard technique for the detection and measurement of proteins, both in the research lab and the clinical setting, it is of great interest to compare our developed biosensor system to the traditional ELISA method.

Our SWCNT-paper biosensor system’s LOD of 2.89 ng/mL is comparable to the LOD of 6.25 ng/mL for the BSA ELISA kit. As our standard curve does not deviate from linearity over its entire range, we suspect that the SWCNT-paper biosensor’s detection range has an upper limit that exceeds 40 ng/mL. In addition, our newly developed method takes no longer than 10 min while analysis by ELISA typically exceeds 4 h.

BSA was utilized in our experiments as a proof-of-concept, in order to determine whether the newly developed biosensor system could detect and measure a standard protein in solution. Overall, our results showed a positive correlation between BSA concentration and biosensor response. Thus, we suspect that our developed biosensors may be applied for the quantification of other proteins, simply by substituting specific antibodies. With the goal of developing a biosensor device that could potentially be deployed in the medical setting for the measurement of human proteins in bodily fluid, we used medically relevant (1–40 ng/mL) BSA concentrations for our experiments. For example, normal levels of prostate specific antigen (PSA), a prostate cancer biomarker in serum, are 0.5 to 2 ng/mL while the danger zone for PSA serum concentration is 4 to 10 ng/mL, a level that is indicative of the earliest stages of prostate cancer [38]. However, many biomarkers in human serum surpass this range, often reaching concentrations of 1–10 µg/mL [39].

Thus, although an increased upper limit of detection for our presented methodology is suspected, further research should be carried out to determine whether it may be applicable for the measurement of protein biomarkers of a broadened range of concentrations.

Due to its medically relevant sensitivity, high specificity and rapid response time, our newly developed method may be more suitable for protein biomarker quantification than the traditional techniques often deployed in the clinical setting, such as ELISA. However, ELISA has a broad range of detection (6.25–400 ng/mL) for the quantification of BSA, and the presented method has only been tested with BSA concentrations within the 1–40 ng/mL range. Although the standard curve for the developed biosensors displayed a strong linear relationship between BSA concentration and biosensor response, and we appropriately suspect that the upper limit of detection may exceed 40 ng/mL, we cannot conclusively compare the detection range of the presented method with ELISA or other relevant methods until further research is carried out.

Additionally, the biosensor system is comprised of cellulose paper and utilizes minimal equipment, making the device more affordable than many standard techniques. The paper-based discs are disposable while the PCB substrate can be reused. In comparison to other rapid, sensitive and portable label-free biosensors for protein quantification that have recently been reported in the literature [16,17,18,19,20,21,22,23,24,25,26,27,28], the presented method has improved cost-effectiveness, simplification and ease of use.

## 4. Conclusions

We describe here an electrical percolation-based label free biosensor capable of quantifying a standard protein. With the goal of meeting the need for a rapid, simplified and affordable analytical device, the biosensors were developed using carbon nanotubes, a specific antibody and cellulose filtration paper. Based on the electrical-percolation principle, the biosensors were hypothesized to sense analytes on the basis of antibody-antigen complex formation, leading to a disruption of the SWCNT-antibody network’s continuity and an increase in the network’s resistance. The developed biosensors comprised an antibody specific to a standard protein, BSA, and were tested with various concentrations of this protein in order to evaluate our hypothesis. Our results show a positive correlation between BSA concentration and biosensor response. The detection limit of the newly developed biosensors was found to be comparable with ELISA. Additionally, the time necessary for detection was found to transcend that of ELISA, and the cost-effectiveness, ease of use and simplicity surpasses those of recent label-free biosensor developments. Further research needs to be carried out in order to conclusively compare the range of detection of the presented method with that of ELISA or other traditional methods. Importantly, the ability to substitute the specific antibody incorporated in the biosensor makes this method incredibly versatile. Due to the high sensitivity, specificity, affordability and rapidity of this device, we believe that it will find wide applications in both biological research and clinical diagnostics.

## Figures and Tables

**Figure 1 biosensors-09-00144-f001:**
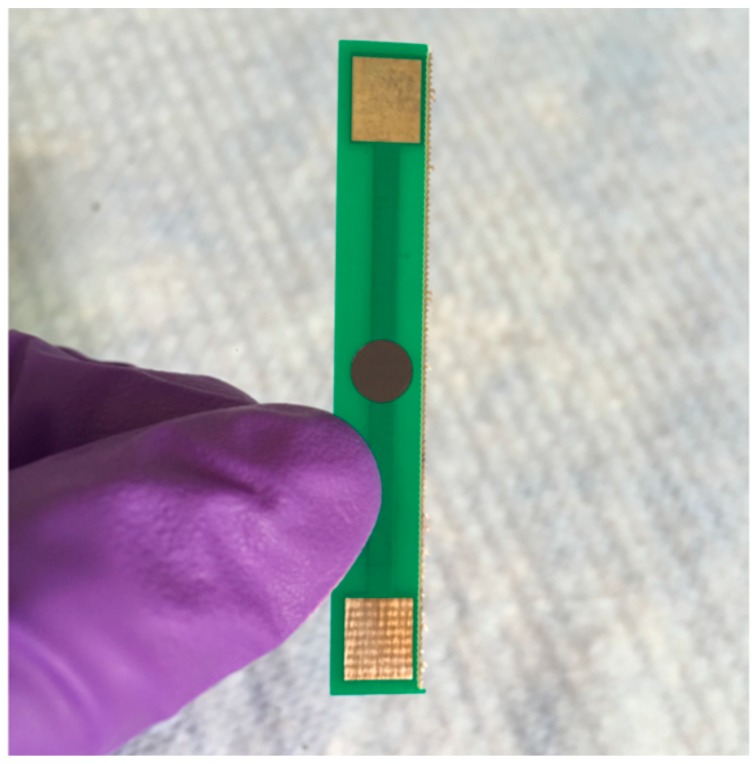
Photograph of a developed biosensor. The image displays the SWCNT-coated paper biosensor disc at the centre of the PCB substrate.

**Figure 2 biosensors-09-00144-f002:**
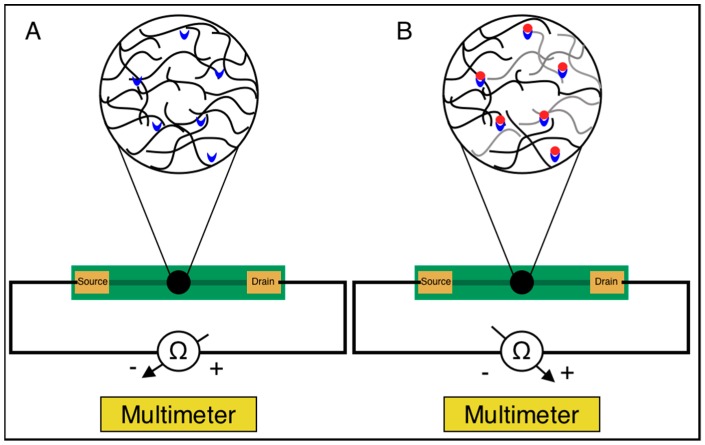
Sensing mechanism of the developed biosensors predicated on the electrical percolation principle with (**A**) The biosensor’s SWCNT (**black**) and antibody (**blue**) network prior to the addition analyte and (**B**) the biosensor’s SWCNT-antibody network following the addition of analyte (**red**), leading to an increase in resistance. Gray SWCNTs are those that are no longer connected in a conductive SWCNT pathway due to the presence of analyte.

**Figure 3 biosensors-09-00144-f003:**
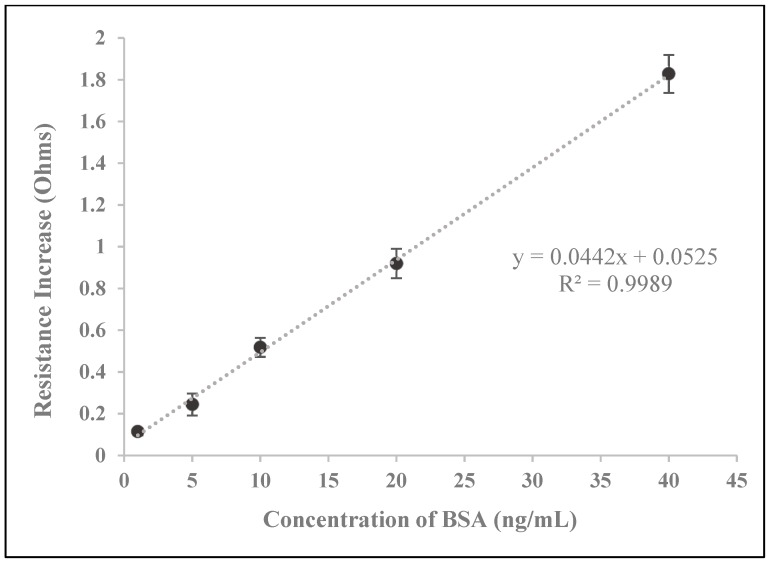
Biosensor detection results. This standard curve resulted from testing the developed biosensors with varying concentrations of BSA.

**Figure 4 biosensors-09-00144-f004:**
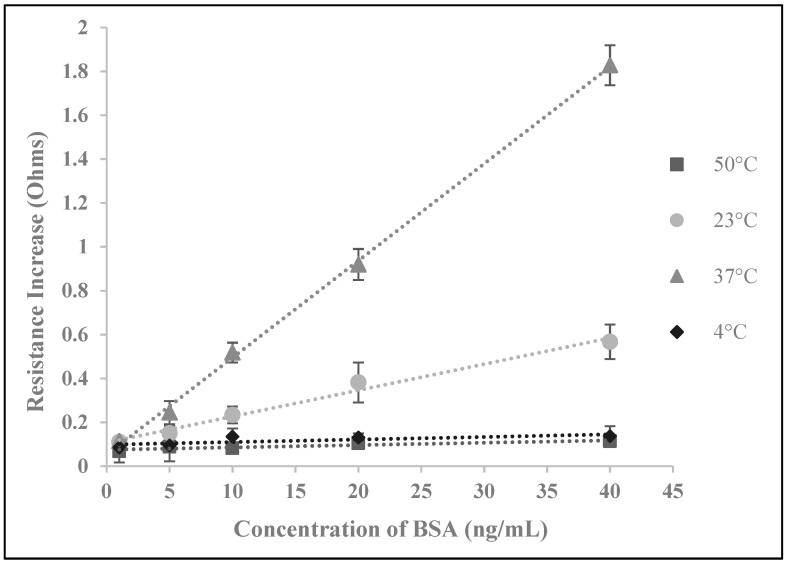
Temperature optimization of the sensing conditions for BSA. Biosensors were tested with various concentrations of BSA at 4 different temperatures.

**Figure 5 biosensors-09-00144-f005:**
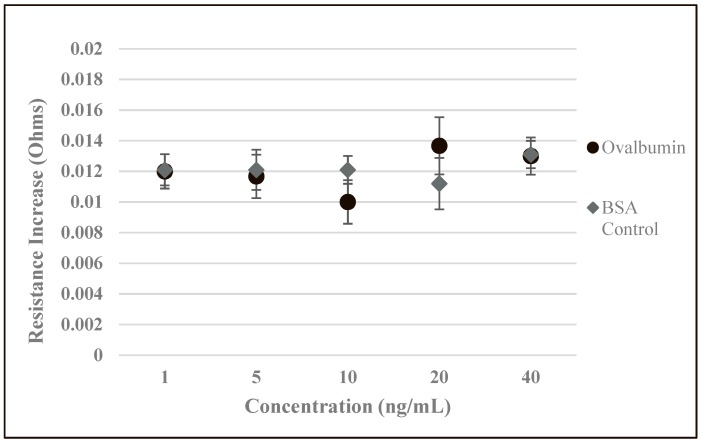
Control experiments with the developed biosensors. Developed biosensors were tested with various concentrations of a non-target analyte, ovalbumin. Biosensors were also developed without the use of BSA antibody and tested with various concentrations of BSA.

**Figure 6 biosensors-09-00144-f006:**
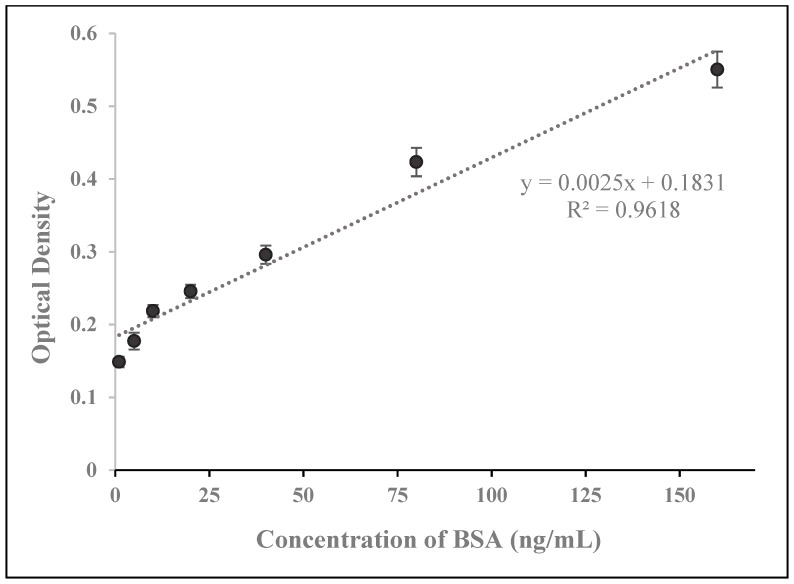
ELISA results for the prepared concentrations of BSA. The various concentrations of BSA that were used to analyze biosensor detection and performance were tested using ELISA for validation.

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
