# Peer review of "Carbon Nanotube-Based Electrochemical Biosensor for Label-Free Protein Detection"

_biosensors, 2019, doi:10.3390/bios9040144_

Round 1

Reviewer 1 Report

The authors of the current manuscript demonstrate label-free sensing of biomolecules, in particular Bovine Serum Albumin, using carbon nanotube-based electrochemical sensor. 

The authors are requested to address the comments below.

The introduction section highlights the salient features of CNTs-based sensors alone. There are many other label-free biosensing modalities, especially those based on optical sensors reported in the literature. Providing some insights with regard to how CNTs-based biosensors compare with other label-free biosensors will help the readers. Authors mention that the antibodies are adsorbed to CNTs via electrostatic interactions mediated by PSS. Did authors conduct any experiments to optimize the ratio of CNTs and PSS? Typically while developing an assay or biosensor, it is important to have a blocking step after antibody immobilization to prevent any non-specific binding of biomolecules to the device itself in subsequent steps. As there is no mention of any blocking step, have the authors tested the condition, wherein, only BSA was added to the device in the absence of any antibody to demonstrate that the biosensor is specific to the target analyte? The authors should provide the supplementary information for the ovalbumin data mentioned on pages 6 and 7.  Authors wait only 5 minutes after the addition of BSA sample to device before conducting measurements. There is no data reported in the manuscript to suggests that the reaction reaches equilibrium within 5 mins. 

Author Response

Dear Reviewer,

We thank you kindly for your helpful criticisms and suggestions. 

1) Explain other label-free biosensing modalities, especially those on optical sensors reported in the literature.

We added a review of other label-free modalities in our introduction, including optical, electrochemical and piezoelectric label-free biosensors reported in the literature.

2) Provide insight with regard to how CNT based biosensors compare with other label-free biosensors

We added a detailed review of CNTs, CNT-based biosensors, and their advantages. We also compared our developed method with other label-free biosensors reported in the literature. 

3) Are there any experiments to optimize the ratio of CNTs to PSS?

Unfortunately, we did not carry out such an experiment.

4) There is no blocking step, so have we tested the condition of just adding BSA with the addition of the BSA antibody?

We have performed this experiment and we added the results alongside the ovalbumin data into the manuscript (Figure 5).

5) Provide supplementary information for the ovalbumin data.

We ended up adding the ovalbumin data to the manuscript (Figure 5)

6) No data in the manuscript to suggest the the reaction reached equilibrium in 5 minutes.

We actually meant that the sensors were allowed to dry for 5 minutes until beginning measurement. 5 minutes was arbitrarily chosen, but we found that all sensors were dry after waiting this long. Measurements were recorded at the transient plateau of resistance readings, and we suspect that this plateau is reached when the reaction is complete. We tried to rephrase this section in hopes of making it more clear. 

Reviewer 2 Report

The manuscript "Carbon Nanotube-Based Electrochemical Biosensor for Label-Free Protein Detection" describes a biosensor composed of carbon nanotubes conjugated with antibodies that may allow quantitative electrical readout of protein concentrations. While the technology is intriguing and shows great potential, the manuscript lacks several crucial control experiments and does not discuss other technologies reported in the literature that serve the same purpose, other than ELISA (see major comment 6). The fact that this manuscript in the big field of biomarker sensing has only 11 references displays that the authors choose to ignore the field or do not know about it. Rather than objectively discussing the limitations and advatages of the here presented techology, the authors praise their own development with words such as 'superb'. Over all, the manuscript does not adhere to scientific standards.

Major comments:

1) The indroduction rarely introduces the reader to the topic but praises CNTs by throwing buzzwords at the reader without explanations. See specific comments 1-13. An example how to improve the introduction is given in line 62-68. This paragraph is explaining the phenomenon of electrical percolation well.

2) The experiments are missing essential controls:

An experiment with the CNT-coated paper without antibodies to show that the readout is    specific to the antibody. It is possible that BSA just sticks to the CNTs or cellulose non-         specifically and that this leads to an increase in resistance. Show the data with Ovalbumin in the same graph as BSA so that the reader can evaluate what you mean with 'slight increase in resistance'.

3) Connecting data points with curved lines is scientifically incorrect as it suggests that data points between would follow these curves. Connecting the dots is unnecessary especially as it obscures the linear fit. Please do not connect discrete data points with lines unless it serves a scientific purpose.

4) Section "3.4. Biosensor Evaluation" is not an evaluation. It is a praise of the here presented technology.

5) Section "3.5. Analysis and Comparison with Traditional Methods" only shows one ELISA experiment, not "methods", plural.

6) The protein concentration range was only tested up to 40ng/mL but many biomarkers are present at much higher concentrations. Please discuss the range of biomarkers as reported in human body fluid such as human serum in respect to the here tested range. For reference see:

- Kuang et al. 2018 Quantitative screening of serum protein biomarkers by reverse phase protein arrays

- Shah et al. 2014 Calibration-free concentration analysis of protein biomarkers in human serum using surface plasmon resonance.

7) The here presented technology must be discussed in the context of existing technologies that serve the same purpose. See point 6) for two examples. Clearly, the here presented CNT-methods has advantageous and if it is only that it is much cheaper and easier to use.

8) An important scientific standard is that experiments have to be described such that other labs can reproduce them. Please clearly describe how you performed your experiments. How was the Arduino Uno programmed and provide the program code.

Specific comments:

9) When an abbreviation is mentioned first in the main text, please explain it. Examples: CNT, LOD.

10) Line 52-53 "great promise for biosensing applications due to a unique combination of electrical, magnetic, optical, mechanical and chemical properties"

- Why? Please explain.

11) Line 55-56 The large surface area of SWCNTs enable increased sensitivity for the detection of biomolecular interactions"

- Why? Please explain.

12) Line 57-58: What do fast electron transfer kinetics have to do with enhanced electrochemical signals and ultrasensitive detection? Please explain.

13) Line 60-61 "Label-free biosensors enable simple real-time detection without the need for secondary

detection elements"

- How? Please explain.

14) Line 151: This is not shown in Fig 2 but in Fig 3.

15) Please be consistent in use of abbreviations, sometimes you use SWCNTs, sometimes CNTs for the same thing.

16) Figure 2: Why are some CNTs gray, others black?

17) Line 218-220: That's the same thing said in both sentences. Taken the range of the ELISA and the actual range of biomarkers in human serum, the half lower detection limit of the presented method is hardly relevant. Furthermore, no statistical analysis was presented to show how the LOD was derived.

18) Line 223: Wrong statement. ELISA does not take 14 hours but only 4 hours: https://www.thermofisher.com/us/en/home/references/protocols/cell-and-tissue-analysis/elisa-protocol/general-elisa-protocol.html

19) Line 232: "superb". Please adhere to scientific language and avoid terms more commonly found in tabloids.

Author Response

Dear Reviewer,

We thank you kindly for your helpful criticisms and suggestions. We also thank you for taking the time to provide such a detailed review and for giving us a chance to improve our manuscript. Here is a point-by-point response to your comments. 

1) The introduction rarely introduces the reader to the topic but praises CNTs by throwing buzzwords at the reader without explanations. See specific comments 1-13. An example how to improve the introduction is given in line 62-68. This paragraph is explaining the phenomenon of electrical percolation well.

We made improvements to our introduction by introducing other label-free biosensor developments for protein quantification reported in the literature. We also added a detailed explanation of carbon nanotubes and SWCNT properties. We explained the advantages of SWCNTs for label-free biosensing. 

2) The experiments are missing essential controls:

An experiment with the CNT-coated paper without antibodies to show that the readout is    specific to the antibody. It is possible that BSA just sticks to the CNTs or cellulose non-         specifically and that this leads to an increase in resistance. Show the data with Ovalbumin in the same graph as BSA so that the reader can evaluate what you mean with 'slight increase in resistance'.

To address this problem, we developed biosensors without the use of BSA antibody and tested the biosensors with various concentrations of BSA. These results were added to the manuscript alongside the ovalbumin experiment results (Figure 5). 

3) Connecting data points with curved lines is scientifically incorrect as it suggests that data points between would follow these curves. Connecting the dots is unnecessary especially as it obscures the linear fit. Please do not connect discrete data points with lines unless it serves a scientific purpose.

We have removed the curved lines connecting our data points. 

4) Section "3.4. Biosensor Evaluation" is not an evaluation. It is a praise of the here presented technology.

We changed the name of this section to "Analysis of Biosensor Performance and Controlled Experiments". 

5) Section "3.5. Analysis and Comparison with Traditional Methods" only shows one ELISA experiment, not "methods", plural.

We changed the name of this section to "Analysis and Comparison with ELISA and Recently Developed Methods" as we added a comparison with recent label-free biosensor developments that have been reported in the literature. 

6) The protein concentration range was only tested up to 40ng/mL but many biomarkers are present at much higher concentrations. Please discuss the range of biomarkers as reported in human body fluid such as human serum in respect to the here tested range. For reference see:

- Kuang et al. 2018 Quantitative screening of serum protein biomarkers by reverse phase protein arrays

- Shah et al. 2014 Calibration-free concentration analysis of protein biomarkers in human serum using surface plasmon resonance.

Thank you for provided these helpful references. We ended up citing the first reference and mentioned that biomarker concentrations tend to surpass 40 ng/ml. We have now stated that further research needs to be carried out in order to determine whether our method is applicable for the quantification of protein biomarkers of a broadened concentration range. 

7) The here presented technology must be discussed in the context of existing technologies that serve the same purpose. See point 6) for two examples. Clearly, the here presented CNT-methods has advantageous and if it is only that it is much cheaper and easier to use.

In our introduction, we have now added a review of other label-free biosensors that have been developed for the same purpose. We also now compare the presented method with these developed technologies reported in the literature in section 3.5. 

8) An important scientific standard is that experiments have to be described such that other labs can reproduce them. Please clearly describe how you performed your experiments. How was the Arduino Uno programmed and provide the program code.

We have now added our source code to our supplementary materials. 

Specific comments:

9) When an abbreviation is mentioned first in the main text, please explain it. Examples: CNT, LOD.

We added an explanation for CNTs, SWCNTs and LOD when first mentioned.

10) Line 52-53 "great promise for biosensing applications due to a unique combination of electrical, magnetic, optical, mechanical and chemical properties"

Why? Please explain.

We further explained how the properties of SWCNTs make them a promising material for use in label-free biosensors. 

11) Line 55-56 The large surface area of SWCNTs enable increased sensitivity for the detection of biomolecular interactions"

Why? Please explain.

To explain this statement, the sentence was changed to: "The large surface area of SWCNTs enables interactions with a large number of biological agents at the carbon nanotube surface, allowing for sensitive detection of biomolecular interactions [33]."

12) Line 57-58: What do fast electron transfer kinetics have to do with enhanced electrochemical signals and ultrasensitive detection? Please explain.

We removed this sentence. 

13) Line 60-61 "Label-free biosensors enable simple real-time detection without the need for secondary

detection elements"

How? Please explain.

We removed this sentence as we have now explain label-free biosensors in the introduction. 

14) Line 151: This is not shown in Fig 2 but in Fig 3.

This was actually meant to be Figure 2. We changed the sentence to make it more clear. We wanted to make reference to our schematic shown in Figure 2. 

15) Please be consistent in use of abbreviations, sometimes you use SWCNTs, sometimes CNTs for the same thing.

We changed all CNT abbreviations to SWCNTs. 

16) Figure 2: Why are some CNTs gray, others black?

We have now added an explanation of the gray and black CNTs below the figure. 

17) Line 218-220: That's the same thing said in both sentences. Taken the range of the ELISA and the actual range of biomarkers in human serum, the half lower detection limit of the presented method is hardly relevant. Furthermore, no statistical analysis was presented to show how the LOD was derived.

We deleted the second sentence. We no longer state that our biosensors exceed the performance of ELISA in terms of LOD. We now state that the LOD is comparable to that of ELISA. We explain how we calculated the LOD on line 147.

18) Line 223: Wrong statement. ELISA does not take 14 hours but only 4 hours: https://www.thermofisher.com/us/en/home/references/protocols/cell-and-tissue-analysis/elisa-protocol/general-elisa-protocol.html

Thank you for catching our mistake. We have now corrected this. 

19) Line 232: "superb". Please adhere to scientific language and avoid terms more commonly found in tabloids.

This word was removed and instead, we use "positive correlation" 

Round 2

Reviewer 1 Report

Thank you for addressing all the comments thoroughly.

Reviewer 2 Report

I would like to thank the authors for addressing all comments appropriately. The manuscript is much improved now. Good luck with the further development of your technology and application on real body fluids.